# Understanding the Effect of Model Compression on Social Bias in Large Language Models

**Gustavo Gonçalves**[1,2] and **Emma Strubell**[1,3]

[1]Language Technologies Institute, Carnegie Mellon University, Pittsburgh, PA, USA
[2]NOVA LINCS, Universidade NOVA de Lisboa, Lisbon, Portugal
[3]Allen Institute for Artificial Intelligence, Seattle, WA, USA
{ggoncalv, estrubel}@cs.cmu.edu

## Abstract

Large Language Models (LLMs) trained with self-supervision on vast corpora of web text fit to the social biases of that text. Without intervention, these social biases persist in the model's predictions in downstream tasks, leading to representational harm. Many strategies have been proposed to mitigate the effects of inappropriate social biases learned during pretraining. Simultaneously, methods for model compression have become increasingly popular to reduce the computational burden of LLMs. Despite the popularity and need for both approaches, little work has been done to explore the interplay between these two. We perform a carefully controlled study of the impact of model compression via quantization and knowledge distillation on measures of social bias in LLMs. Longer pretraining and larger models led to higher social bias, and quantization showed a regularizer effect with its best trade-off around 20% of the original pretraining time. [1]

## 1 Introduction

Large Language Models (LLMs) are trained on large corpora using self-supervision, which allows models to consider vast amounts of unlabelled data, and learn language patterns through masking tasks (Devlin et al., 2019; Radford et al., 2019). However, self-supervision allows LLMs to pick up social biases contained in the training data. Which is amplified by larger models, more data, and longer training (Kaneko et al., 2022; Kaneko and Bollegala, 2022; Kurita et al., 2019; Delobelle and Berendt, 2022).

Social biases in LLMs are an ongoing problem that is propagated from pretraining to finetuning (Ladhak et al., 2023; Gira et al., 2022). Biased pretrained models are hard to fix, as retraining is prohibitively expensive both financially and environmentally (Hessenthaler et al., 2022). At the same time, the compression of LLMs has been intensely studied. Pruning, quantization, and distillation are among the most common strategies to compress LLMs. Pruning reduces the parameters of a trained model by removing redundant connections while preserving equivalent performance to their original counterparts (Liebenwein et al., 2021; Ahia et al., 2021). Quantization reduces the precision of model weights and activations to improve efficiency while preserving performance (Ahmadian et al., 2023). Finally, knowledge distillation (Hinton et al., 2015) trains a smaller more efficient model based on a larger pre-trained model.

While much research has been done on measuring and mitigating social bias in LLMs, and making LLMs smaller and more efficient, by using one or a combination of many compression methods (Xu et al., 2021), little research has been done regarding the interplay between social biases and LLM compression. Existing work has shown that pruning disproportionately impacts classification accuracy on low-frequency categories in computer vision models (Hooker et al., 2021), but that pruning transformer models can have a beneficial effect with respect to bias when modeling multilingual text (Hooker et al., 2020; Ogueji et al., 2022). Further, Xu and Hu (2022) have shown that compressing pretrained models improves model fairness by working as a regularizer against toxicity.

Unlike previous work, our work focuses on the impacts of widely used quantization and distillation on the social biases exhibited by a variety of both encoder- and decoder-only LLMs. We focus on the effects of social bias over BERT (Devlin et al., 2019), RoBERTa (Liu et al., 2019) and Pythia LLMs (Biderman et al., 2023). We evaluate these models against Bias Bench (Meade et al., 2022), a compilation of three social bias datasets.

In our experimental results we demonstrate a cor-

---

[1]https://github.com/gsgoncalves/EMNLP2023_llm_compression_and_social_bias

relation between longer pretraining, larger models, and increased social bias, and show that quantization and distillation can reduce bias, demonstrating the potential for compression as a pragmatic approach for reducing social bias in LLMs.

## 2 Methodology

We were interested in understanding how dynamic Post-Training Quantization (PTQ) and distillation influence social bias contained in LLMs of different sizes, and along their pretraining. In dynamic PTQ, full-precision floating point model weights are statically mapped to lower precisions after training, with activations dynamically mapped from high to low precision during inference. To this end, in Section 2.1 we present the datasets of the Bias Bench benchmark (Meade et al., 2022) that enable us to evaluate three different language modeling tasks across the three social bias categories. In Section 2.2 we lay out the models we studied. We expand on the Bias Bench original evaluation by looking at the Large versions of the BERT and RoBERTa models, and the Pythia family of autoregressive models. The chosen models cover different language modeling tasks and span across a wide range of parameter sizes, thus providing a comprehensive view of the variations of social bias.

### 2.1 Measuring Bias

We use the Bias Bench benchmark for evaluating markers of social bias in LLMs. Bias Bench compiles three datasets, CrowS-Pairs (Nangia et al., 2020), StereoSet (SS) (Nadeem et al., 2021), and SEAT (Kaneko and Bollegala, 2021), for measuring intrinsic bias across three different identity categories: GENDER, RACE, and RELIGION. While the set of identities covered by this dataset is far from complete, it serves as a useful indicator as these models are encoding common social biases; however, the lack of bias indicated by this benchmark does not imply an overall lack of inappropriate bias in the model, for example with respect to other groups. We briefly describe each dataset below; refer to the original works for more detail.

**CrowS-Pairs** is composed of pairs of minimally distant sentences that have been crowdsourced. A minimally distant sentence is defined as a small number of token swaps in a sentence, that carry different social bias interpretations. An unbiased

model will pick an equal ratio of both stereotypical and anti-stereotypical choices, thus an optimal score for this dataset is a ratio of 50%.

**StereoSet** is composed of crowdsourced samples. Each sample is composed of a masked context sentence, and a set of three candidate answers: 1) stereotypical, 2) anti-stereotypical, and 3) unrelated. Under the SS formulation, an unbiased model would give a balanced number of classifications of types 1) and 2), thus the optimal score is also 50%. The SS dataset also measures if we are changing the language modeling properties of our model. That is, if our model picks a high percentage of unrelated choices 3) it can be interpreted as losing its language capabilities. This is defined as the Language Model (LM) Score.

**SEAT** evaluates biases in sentences. A SEAT task is defined by two sets of attribute sentences, and two other sets of target sentences. The objective of the task is to measure the distance of the sentence embeddings between the attribute and target sets to assess a preference between attributes and targets (bias). We provide more detail of this formulation in Appendix A.1.

### 2.2 Models

In this work, we focus on two popular methods for model compression: knowledge distillation and quantization. We choose these two methods given their competitive performance, wide deployment given the availability of distributions under the HuggingFace and Pytorch libraries, and the lack of understanding of the impact of these methods on social biases. We leave the study of more elaborate methods for improving model efficiency such as pruning (Chen et al., 2020), mixtures of experts (Kudugunta et al., 2021), and adaptive computation (Elbayad et al., 2020) to future work.

Since model compression affects model size, we are particularly interested in understanding how pretrained model size impacts measures of social bias, and how that changes as a function of how well the model fits the data. We are also interested in investigating how the number of tokens observed during training impacts all of the above. We experiment with three different base LLMs: BERT (Devlin et al., 2019), RoBERTa (Liu et al., 2019), and Pythia (Biderman et al., 2023), with uncompressed model sizes ranging from 70M parameters to 6.9B parameters. BERT and RoBERTa

| Model | Params | Size (MB) | GENDER | RACE | RELIGION |
|---|---|---|---|---|---|
| BERT Base | 110M | 438 | 57.25 | 62.33 | 62.86 |
| + DYNAMIC PTQ int8 | 110M | 181 | 57.25 | ↓0.19 62.14 | ↓9.53 46.67 |
| + CDA (Webster et al., 2020) | 110M | | ↓1.14 56.11 | ↓5.63 56.70 | ↓2.86 60.00 |
| + DROPOUT (Webster et al., 2020) | 110M | | ↓1.91 55.34 | ↓3.30 59.03 | ↓7.62 55.24 |
| + INLP (Ravfogel et al., 2020) | 110M | | ↓6.10 51.15 | ↑5.63 67.96 | ↓1.91 60.95 |
| + SELF-DEBIAS (Schick et al., 2021) | 110M | | ↓4.96 52.29 | ↓5.63 56.70 | ↓6.67 56.19 |
| + SENTDEBIAS (Liang et al., 2020) | 110M | | ↓4.96 52.29 | ↑0.39 62.72 | ↑0.95 63.81 |
| BERT Large | 345M | 1341 | ↓1.52 55.73 | ↓1.94 60.39 | ↑4.76 67.62 |
| + DYNAMIC PTQ int8 | 345M | 432 | ↓6.87 **50.38** | ↑0.78 63.11 | ↓7.62 55.24 |
| DistilBERT | 66M | 268 | ↓6.10 51.15 | ↓9.32 **46.99** | ↓4.76 58.10 |
| RoBERTa Base | 123M | 498 | 60.15 | 63.57 | 60.95 |
| + DYNAMIC PTQ int8 | 123M | 242 | ↓6.51 53.64 | ↓5.04 58.53 | ↓10.47 **49.52** |
| + CDA (Webster et al., 2020) | 110M | | ↓3.83 56.32 | ↑0.19 63.76 | ↓0.95 59.05 |
| + DROPOUT (Webster et al., 2020) | 110M | | ↓0.76 59.39 | ↓1.17 62.40 | ↓2.86 57.14 |
| + INLP (Ravfogel et al., 2020) | 110M | | ↓4.98 55.17 | ↓1.75 61.82 | ↑1.91 62.86 |
| + SELF-DEBIAS (Schick et al., 2021) | 110M | | ↓3.06 57.09 | ↓1.17 62.40 | ↓9.52 51.43 |
| + SENTDEBIAS (Liang et al., 2020) | 110M | | ↓8.04 52.11 | ↑1.55 65.12 | ↓1.9 40.95 |
| RoBERTa Large | 354M | 1422 | 60.15 | ↑0.58 64.15 | ↑0.95 61.90 |
| + DYNAMIC PTQ int8 | 354M | 513 | ↓2.68 57.47 | ↓0.20 63.37 | ↓0.95 60.00 |
| DistilRoBERTa | 82M | 329 | ↓7.28 52.87 | ↓3.49 60.08 | ↑2.86 63.81 |

Table 1: CrowS-Pairs stereotype scores for GENDER, RACE, and RELIGION for BERT and RoBERTa models. Stereotype scores closer to 50% indicate less biased model behavior. Bold values indicate the best method per bias category. Results on the other datasets displayed similar trends and were included in Appendix B for space.

represent two similar sets of widely used and studied pretrained architectures, trained on different data with a small overlap. RoBERTa pretraining was done over 161 GB of text, which contained the 16GB used to train BERT, approximately a ten-fold increase. RoBERTa also trained for longer, with larger batch sizes which have shown to decrease the perplexity of the LLM (Liu et al., 2019).

The set of checkpoints released for the Pythia model family allows us to assess an even wider variety of model sizes and number of training tokens, including intermediate checkpoints saved during pretraining, so that we can observe how bias varies throughout pretraining. We used the models pretrained on the deduplicated version of The Pile (Gao et al., 2021) containing 768GB of text.

**Knowledge distillation** (Hinton et al., 2015) is a popular technique for compressing the knowledge encoded in a larger teacher model into a smaller student model. In this work, we analyze Distil-BERT (Sanh et al., 2019) and DistilRoBERTa[2] distilled LMs. During training the student model minimizes the loss according to the predictions of the teacher model (soft-targets) and the true labels (hard-targets) to better generalize to unseen data.

**Quantization** compresses models by reducing the precision of their weights and activations during inference. We use the standard PyTorch implementation[3] to apply dynamic PTQ over the linear layers of the transformer stack, from fp32 full-precision to quantized int8 precision. This work analyzes quantized BERT, RoBERTa, and Pythia models of a comprehensive range of sizes.

## 3 Results

**Dynamic PTQ and distillation lower social bias.** In Table 1 we analyze the effects of dynamic PTQ and distillation in the CrowS dataset, where BERT Base and RoBERTa Base are our baselines. To compare quantization and distillation, we add three debiasing baselines also referenced by Meade et al. (2022) that are competitive strategies to reduce bias. The INLP (Ravfogel et al., 2020) baseline consists of a linear classifier that learns to predict the target bias group given a set of context words, such as

[2] https://huggingface.co/distilroberta-base

[3] https://pytorch.org/tutorials/recipes/recipes/dynamic_quantization.html

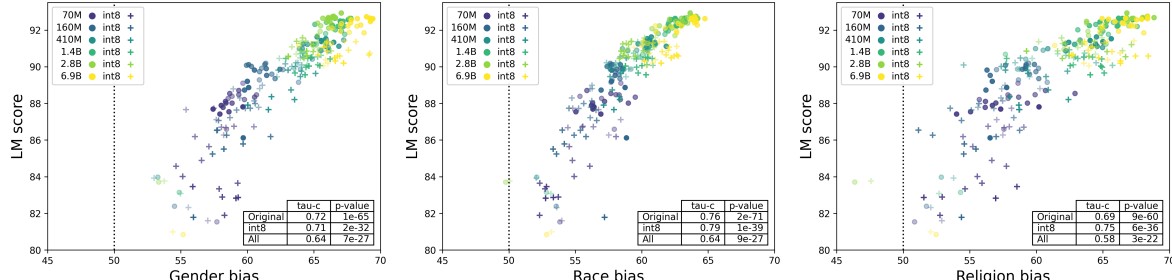

Figure 1: LM score vs. GENDER, RACE, and RELIGION bias on the SS dataset across all Pythia models. Darker data points show later pretraining steps, and more transparent points to earlier steps. The included table shows the Kendall Tau C, for the correlation across "All" model sizes, full-precision "Original", and "int8" model sizes.

| Model Size | Best LM Score | Step Nr. | Bias G. / RA. / RE. |
|---|---|---|---|
| 70M | 89.2 | 21K | 59.8 / 58.4 / 58.6 |
| 160M | 90.2 | 36K | 61.4 / 57.6 / 59.4 |
| 410M | 91.6 | 114K | 65.2 / 60.7 / 64.5 |
| 1.4B | 92.6 | 129K | 66.6 / 63.2 / 66.2 |
| 2.8B | 92.9 | 114K | 67.1 / 63.7 / 66.8 |
| 6.9B | 92.7 | 129K | 69.0 / 64.0 / 68.4 |

Table 2: Bias measured using SS for the full-precision Pythia models having the best LM score per model size.

| Model Size | Best LM Score | Step Nr. | Bias G. / RA. / RE. |
|---|---|---|---|
| 70M | 87.7 | 29K | 57.5 / 54.8 / 58.0 |
| 160M | 89.0 | 21K | 61.1 / 56.3 / 57.7 |
| 410M | 90.5 | 50K | 64.2 / 58.4 / 63.6 |
| 1.4B | 91.4 | 29K | 66.1 / 59.7 / 63.3 |
| 2.8B | 91.6 | 50K | 64.1 / 60.2 / 61.9 |
| 6.9B | 91.4 | 21K | 67.3 / 60.1 / 67.3 |

Table 3: Bias measured using SS for int8 quantized Pythia models having the best LM score per model size.

*'he/she'*. The Self-Debias baseline was proposed by Schick et al. (2021), and uses prompts to encourage models to generate toxic text and learns to give less weight to the generate toxic tokens. Self-Debias does not change the model's internal representation, thus it cannot be evaluated on the SEAT dataset.

Notable trends in Table 1 are the reduction of social biases when applying dynamic PTQ and distillation, which can compete on average with the specifically designed debias methods. Additional results in in Appendix B also display similar trends. On the SS dataset in Table 4 we are also able to observe that the application of distillation provides remarkable decreases in social biases, at the great expense of LM score. However, dynamic PTQ shows a better trade-off in providing social bias reductions, while preserving LM score.

**One model size does not fit all social biases.** In Table 1 and the equivalent Tables in Appendix B we can see that social bias categories respond differently to model size, across the different datasets. While BERT Base/Large outperforms RoBERTa in GENDER, the best model for RACE and RELIGION varies across datasets. This can be explained by the different dataset tasks and the pretraining.

In Appendix B we show the social bias scores as

a function of the pretraining of the Pythia models in Figures 2 to 7, 9, 10 and 11. The BERT/RoBERTa Base and Large versions are roughly comparable with the 160M and 410M Pythia models. For the SS dataset, the 160M model is consistently less biased than the 410M model. However, this is not the case for the other two datasets where the 160M struggles in the RACE category while assessing the distance of sentence embeddings (SEAT); and in the RELIGION category while swapping minimally distant pairs (CrowS). This illustrates the difficulty of distinguishing between semantically close words, and shows the need for larger models pretrained for longer and on more data.

**Longer pretraining and larger models lead to more socially biased models.** We study the effects of longer pretraining and larger models on social bias, by establishing the correlation of these variables in Figure 1. Here we can observe that as the model size increases so does the LM model score and social bias across the SS dataset. Moreover, later stages of pretraining have a higher LM model score, where the social bias score tends to be high. The application of dynamic PTQ shows a regularizer effect on all models. The Kendall Tau C across the models and categories shows a strong

correlation between LM score and social bias. Statistical significant tests were performed using a one-sided t-test to evaluate the positive correlation.

Tables 2 and 3 show at what step, out of the 21 we tested, the best LM scores occur on the SS dataset. In Table 2 the best LM score increases monotonically with model size and so do the social biases. Interestingly, as the model size increases the best LM score appears after around 80% of the pretraining. In opposition, in Table 3, with dynamic PTQ the best LM score occurs around 20% of the pretraining and maintains the trend of higher LM score and social bias, albeit at lower scores than the original models. This shows an interesting possibility of early stopping depending on the deployment task of the LLM.

## 4 Limitations

While this work provides three different datasets, which have different views on social bias and allow for an indicative view of LLMs, they share some limitations that should be considered. The datasets SS and CrowS define an unbiased model as one that makes an equal amount of stereotypical and anti-stereotypical choices. While we agree that this makes a good definition of an impartial model it is a limited definition of an unbiased model. This has also been noted by Blodgett et al. (2021), showing that CrowS is slightly more robust than SS by taking "extra steps to control for varying base rates between groups." (Blodgett et al., 2021). We should consider that these datasets depict mostly Western biases, and the dataset construction since it is based on assessors it is dependent on the assessor's views. Moreover, Blodgett et al. (2021) has also noted the existence of unbalanced stereotype pairs in SS and CrowS, and the fact that some samples in the dataset are not consensual stereotypes.

All datasets only explore three groups of biases: GENDER, RACE, and RELIGION, which are not by any means exhaustive representations of social bias. The experiments in this paper should be considered indicative of social bias and need to be further studied. Additionally, the GENDER category is defined as binary, which we acknowledge that does not reflect the timely social needs of LLMs, but can be extended to include non-binary examples by improving on existing datasets.

We benefited from access to a cluster with two AMD EPYC 7 662 64-Core Processors, where the quantized experiments ran for approximately 4 days. A CPU implementation was used given the quantization backends available in PyTorch. Experiments that did not require quantization ran using an NVIDIA A100 40GB GPU and took approximately 5 hours to run.

## Ethics Statement

We reiterate that this work provides a limited Western view of Social bias focusing only on three main categories: GENDER, RACE, and RELIGION. Our work is further limited to a binary definition of GENDER, which we acknowledge that does not reflect the current society's needs. Moreover, we must also reiterate that these models need to be further studied and are not ready for production. The effects of quantization along pretraining should be considered as preliminary results.

## 5 Acknowledgments

This work has been partially funded by the FCT project NOVA LINCS Ref. UIDP/04516/2020, by the Amazon Science - TaskBot Prize Challenge and the CMU|Portugal projects iFetch Ref. LISBOA-01-0247-FEDER-045920 and GoLocal Ref. CMUP-ERI/TIC/0046/2014, and by the FCT Ph.D. scholarship grant Ref. SFRH/BD/140924/2018. We would like to acknowledge the NOVASearch group for providing compute resources for this work. Any opinions, findings, and conclusions in this paper are the authors' and do not necessarily reflect those of the sponsors.

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

## A Details of Metric Calculation

### A.1 SEAT

The SEAT task shares the same task as WEAT task, which is defined by four word sets, two attribute sets, and two target sets. For example, to decide the presence of gender bias the two attribute sets are disjoint sets given by: 1) a masculine set of words, such as *{'man', 'boy', 'he', ...}*, and 2) a set of feminine words *{'woman', 'girl', 'her', ...}*. The target sets will characterize concepts such as 'sports' and 'culinary'.

WEAT evaluates how close are the attribute sets from the target sets to determine the existence of bias. Mathematically this is given by:

$$s(A, B, X, Y) = \sum_{x \in X} s(x, A, B) - \sum_{y \in Y} s(y, A, B)$$

(1)

Where $A$ and $B$ represent the attribute sets, and $X$ and $Y$ are the target sets of words. The $s$ function in Equation (1) denotes mean cosine similarity between the target word embeddings and the attribute word embeddings:

$$s(w, A, B) = \frac{1}{|A|} \sum_{a \in A} \cos(w, a) - \frac{1}{|B|} \sum_{b \in B} \cos(w, b).$$

(2)

The reported score of the benchmark (effect size) is given by:

$$d = \frac{\mu(\{s(x, A, B)\}_{x \in X}) - \mu(\{s(y, A, B)\}_{y \in Y})}{\sigma(\{s(t, X, Y)\}_{t \in A \cup B})}$$

(3)

Where $\mu$ and $\sigma$ are the mean and standard deviation respectively. Equation (3) is designed so that scores closer to zero indicate the smallest possible degree of bias. SEAT extends the previous formulation by considering the distance sentence embeddings instead of word embeddings.

## B Additional Plots and Tables

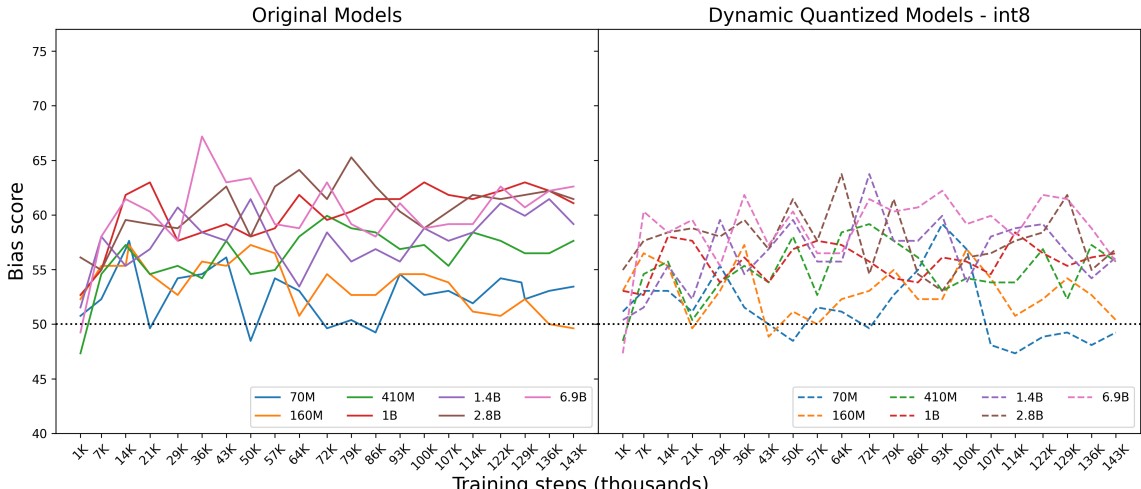

Figure 2: Crows GENDER bias with Quantized Results

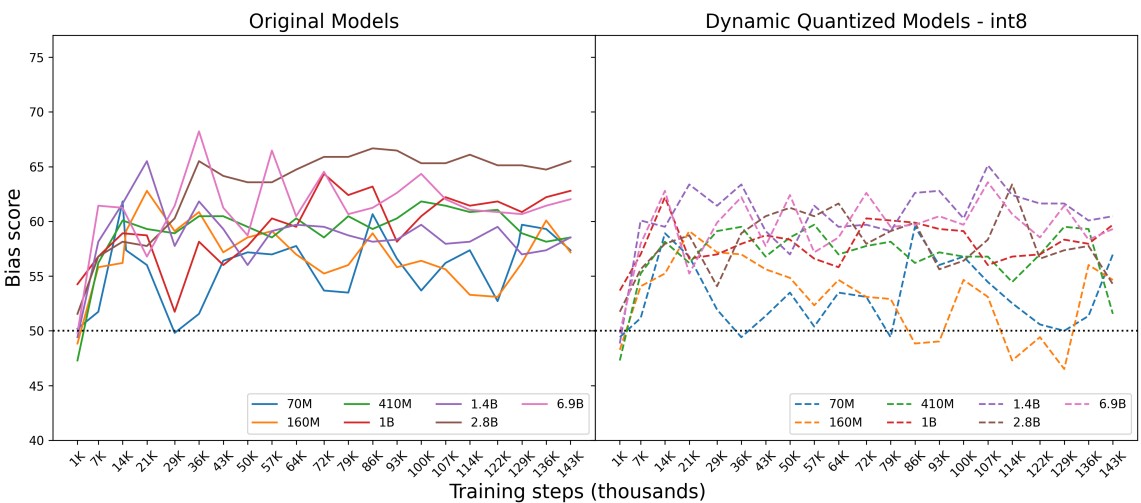

Figure 3: Crows RACE bias with Quantized Results

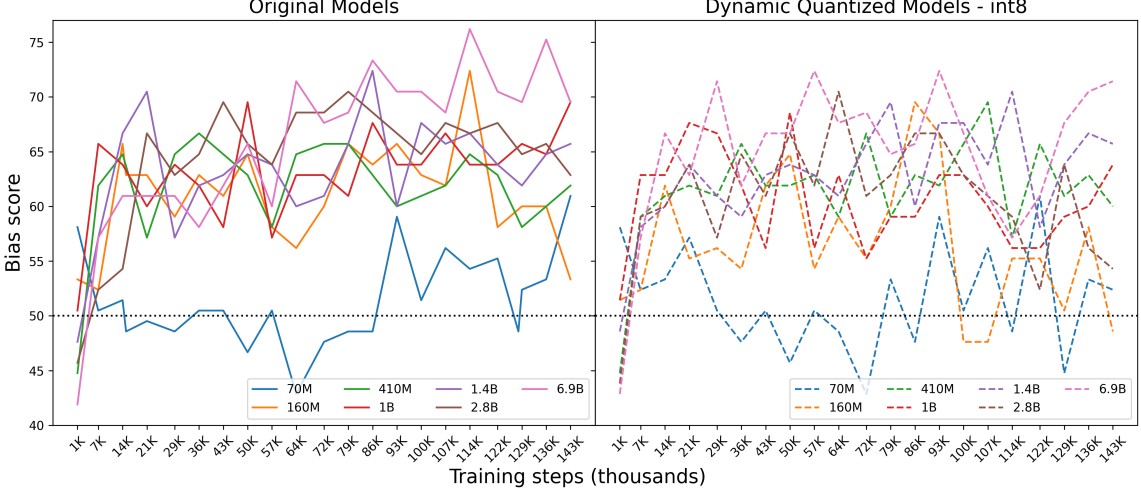

Figure 4: Crows RELIGION bias with Quantized Results

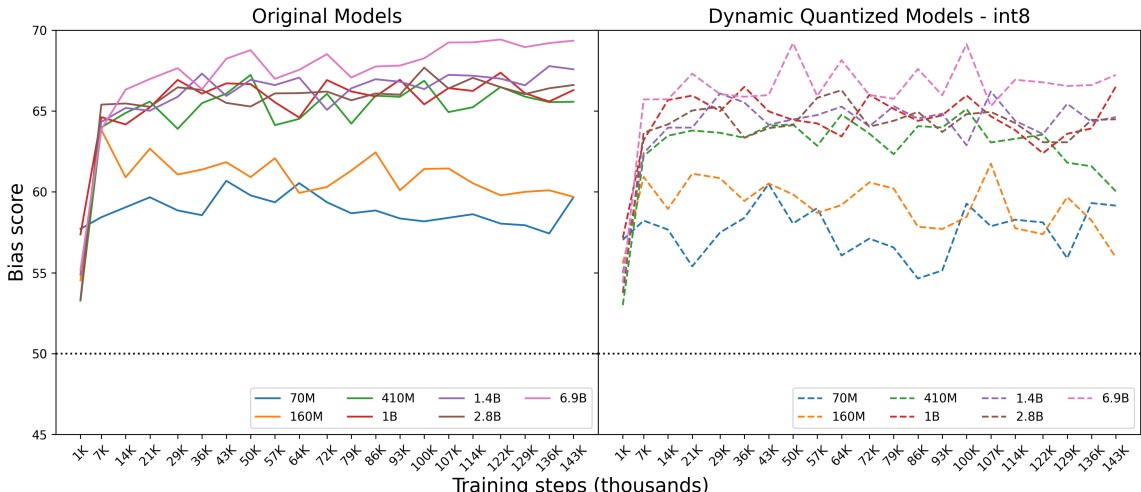

Figure 5: Stereoset GENDER bias with Quantized Results

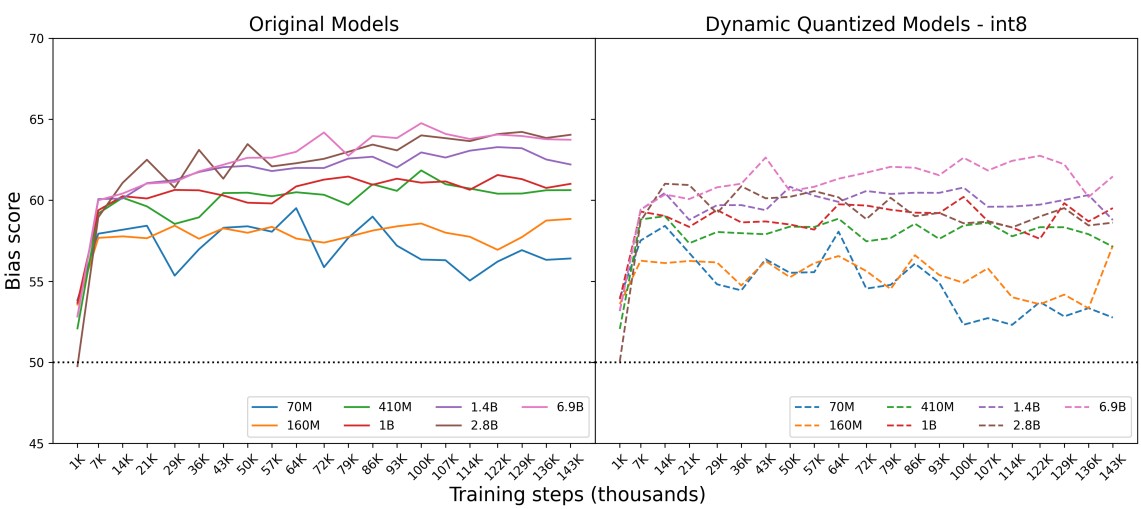

Figure 6: Stereoset RACE bias with Quantized Results

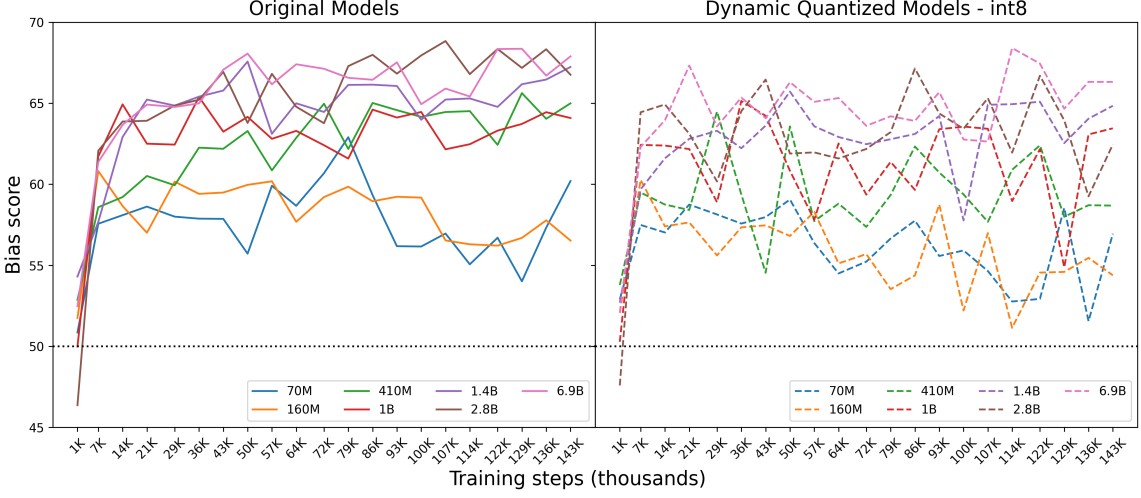

Figure 7: Stereoset RELIGION bias with Quantized Results

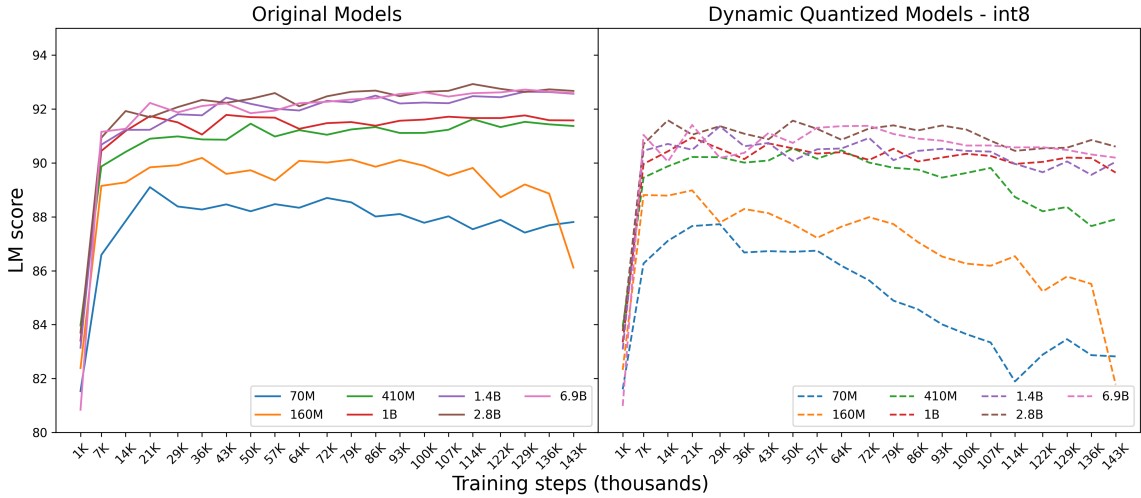

Figure 8: Stereoset LM Score with Quantized Results

Table 4: SS stereotype scores and language modeling scores (LM Score) for BERT, and RoBERTa models. Stereotype scores closer to 50% indicate less biased model behavior. Bold values indicate the best method per bias and LM Score. Results are on the SS test set. A random model (which chooses the stereotypical candidate and the anti-stereotypical candidate for each example with equal probability) obtains a stereotype score of 50% in expectation.

| Model | GENDER bias | RACE bias | RELIGION bias | LM Score |
|---|---|---|---|---|
| BERT Base | 60.28 | 57.03 | 59.70 | 84.17 |
| + DYNAMIC PTQ int8 | ↓3.29 56.99 | ↓2.36 54.67 | ↓2.87 56.83 | ↓2.94 81.23 |
| + CDA (Webster et al., 2020) | ↓0.67 59.61 | ↓0.30 56.73 | ↓1.33 58.37 | ↓1.09 83.08 |
| + DROPOUT (Webster et al., 2020) | ↑0.38 60.66 | ↑0.04 57.07 | ↓0.57 59.13 | ↓1.14 83.04 |
| + INLP (Ravfogel et al., 2020) | ↓3.03 57.25 | ↑0.26 57.29 | ↓2.44 57.26 | ↓3.54 80.63 |
| + SELF-DEBIAS (Schick et al., 2021) | ↓0.94 59.34 | ↓2.73 54.30 | ↓2.44 57.26 | ↓0.08 84.09 |
| + SENTENCEDEBIAS (Liang et al., 2020) | ↓0.91 59.37 | ↑0.75 57.78 | ↓0.97 58.73 | ↑0.03 84.20 |
| BERT Large | ↑2.96 63.24 | ↑0.04 57.07 | ↑0.24 59.94 | ↑0.24 84.41 |
| + DYNAMIC PTQ int8 | ↓0.82 59.46 | ↓1.86 55.17 | ↓3.74 55.96 | ↓3.12 81.05 |
| Distil BERT Base | ↓8.73 **51.55** | ↓6.40 **50.63** | ↓9.57 **49.87** | ↓30.30 53.87 |
| RoBERTa Base | 66.32 | 61.67 | 64.28 | 88.95 |
| + DYNAMIC PTQ int8 | ↓3.92 62.40 | ↓3.15 58.52 | ↓0.03 64.25 | ↓5.75 83.20 |
| + CDA (Webster et al., 2020) | ↓1.89 64.43 | ↓0.73 60.95 | ↓0.23 64.51 | ↓0.10 83.83 |
| + DROPOUT (Webster et al., 2020) | ↓0.06 66.26 | ↓1.27 60.41 | ↓2.20 62.08 | ↓0.11 88.81 |
| + INLP (Ravfogel et al., 2020) | ↓9.06 60.82 | ↓3.41 58.26 | ↓3.94 60.34 | ↓0.70 88.23 |
| + SELF-DEBIAS (Schick et al., 2021) | ↓1.28 65.04 | ↓2.89 58.78 | ↓1.44 62.84 | ↓0.67 88.26 |
| + SENTENCEDEBIAS (Liang et al., 2020) | ↓3.55 62.77 | ↑1.05 62.72 | ↓0.37 63.91 | ↑0.01 88.94 |
| RoBERTa Large | ↑0.51 66.83 | ↓1.37 60.30 | ↑0.21 64.49 | ↑0.14 89.09 |
| + DYNAMIC PTQ int8 | ↓2.72 63.60 | ↓2.10 59.57 | ↓0.40 63.88 | ↓0.68 88.27 |
| Distil RoBERTa Base | ↓2.04 64.28 | ↓0.36 61.31 | ↑1.16 65.44 | ↑0.24 **89.19** |

Table 5: LM Scores vs. Biases on the SS dataset of the int8 models, at the same steps with the best LM Score for the original (full-precision) models (Table 2)

| Model Size | LM Score | Step Nr. | Bias G. / RA. / RE. |
|---|---|---|---|
| 70M | 87.7 | 21K | 55.4 / 56.8 / 58.8 |
| 160M | 88.3 | 36K | 59.4 / 54.7 / 57.3 |
| 410M | 88.7 | 114K | 63.3 / 57.8 / 60.9 |
| 1.4B | 90.1 | 129K | 65.5 / 60.0 / 62.5 |
| 2.8B | 90.5 | 114K | 64.3 / 58.3 / 62.0 |
| 6.9B | 90.5 | 129K | 66.6 / 62.2 / 64.7 |

Table 6: LM Scores vs. Biases on the SS dataset of the original (full-precision) models, at the same steps with the best LM Score for the int8 models (Table 3)

| Model Size | LM Score | Step Nr. | Bias G. / RA. / RE. |
|---|---|---|---|
| 70M | 88.4 | 29K | 58.9 / 55.4 / 58.0 |
| 160M | 89.8 | 21K | 62.7 / 57.7 / 57.0 |
| 410M | 91.5 | 50K | 67.2 / 60.5 / 63.3 |
| 1.4B | 91.8 | 29K | 65.9 / 61.2 / 64.9 |
| 2.8B | 92.4 | 50K | 65.3 / 63.5 / 63.8 |
| 6.9B | 92.2 | 21K | 67.0 / 61.0 / 64.9 |

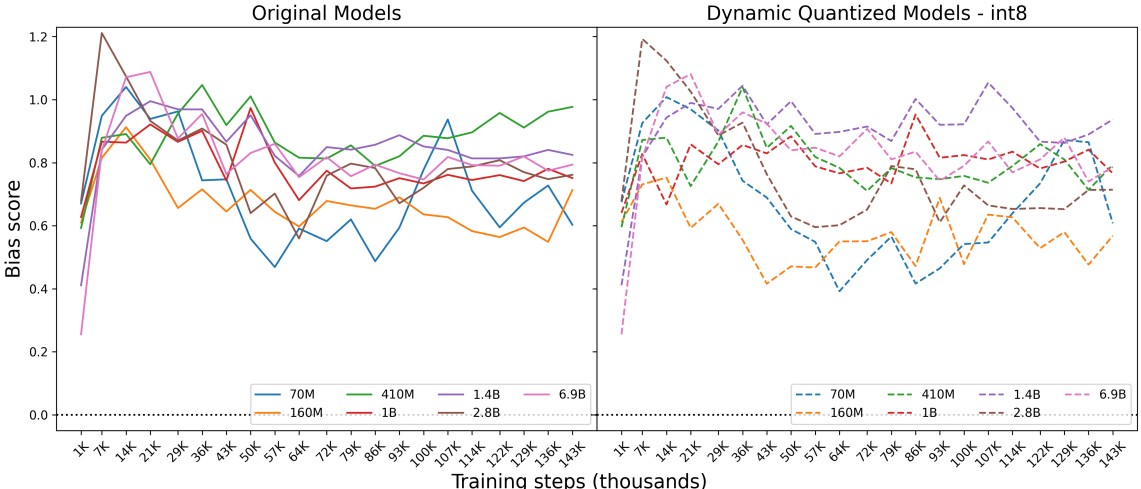

Figure 9: Seat GENDER bias with Quantized Results

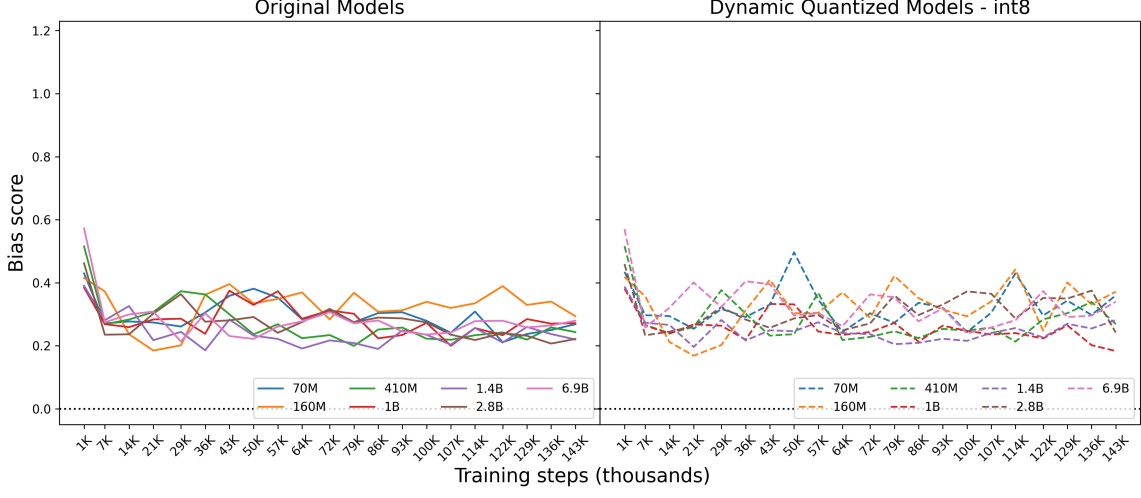

Figure 10: Seat RACE bias with Quantized Results

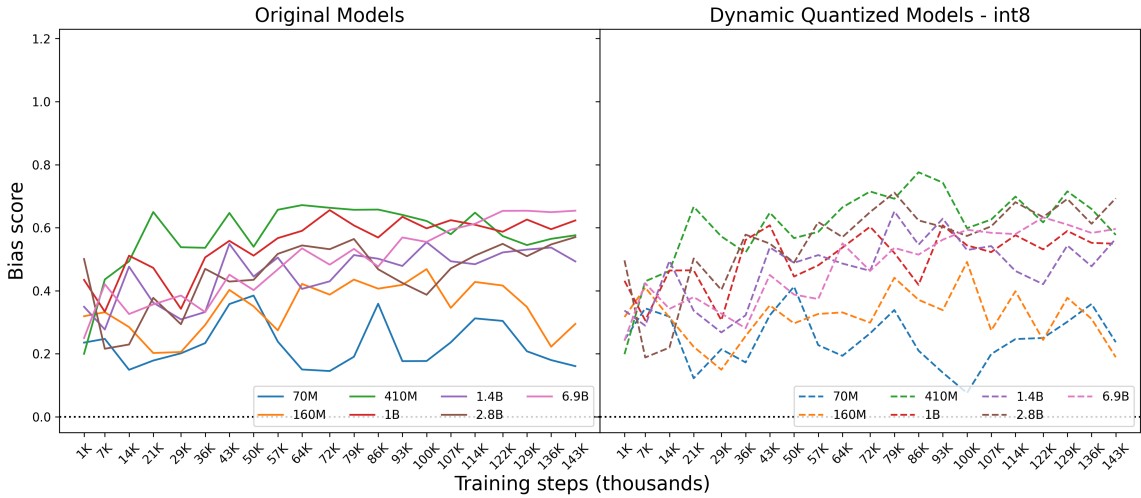

Figure 11: Seat RELIGION bias with Quantized Results

Table 7: GENDER bias on SEAT dataset. Effect sizes closer to 0 are indicative of less biased model representations. Bold values indicate the best method per test. Statistically significant effect sizes at p < 0.01 are denoted by *. The final column reports the average absolute effect size across all six gender SEAT tests for each model.

| Model | weat6 | weat6b | weat7 | weat7b | weat8 | weat8b | Avg. Effect | |
|---|---|---|---|---|---|---|---|---|
| BERT Base | 0.931 * | 0.090 | -0.124 | 0.937 * | 0.783 * | 0.858 * | | 0.620 |
| + DYNAMIC PTQ int8 | 0.614 * | **0.000** | -0.496 | 0.711 * | 0.401 | 0.549 * | ↓0.158 | 0.462 |
| + CDA | 0.846 * | 0.186 | -0.278 | 1.342 * | 0.831 * | 0.849 * | ↑0.102 | 0.722 |
| + DROPOUT | 1.136 * | 0.317 | 0.138 | 1.179 * | 0.879 * | 0.939 * | ↑0.144 | 0.765 |
| + INLP | 0.317 | -0.354 | -0.258 | **0.105** | 0.187 | **-0.004** | ↓0.416 | 0.204 |
| + SENTENCEDEBIAS | 0.350 | -0.298 | -0.626 | 0.458 * | 0.413 | 0.462 * | ↓0.186 | 0.434 |
| BERT Large | 0.370 | -0.015 | 0.418 * | 0.221 | -0.259 | 0.710 * | ↓0.288 | 0.332 |
| + DYNAMIC PTQ int8 | 0.905 * | 0.273 | 1.097 * | 0.894 * | 0.728 * | 1.180 * | ↑0.226 | 0.846 |
| Distil BERT | **0.061** | -0.222 | **0.093** | -0.120 | 0.222 | 0.112 | ↓0.482 | **0.138** |
| RoBERTa Base | 0.922 * | 0.208 | 0.979 * | 1.460 * | 0.810 * | 1.261 * | | 0.940 |
| + DYNAMIC PTQ int8 | 0.350 | 0.177 | 0.389 * | 1.038 * | 0.349 | 0.897 * | ↓0.406 | 0.533 |
| + CDA | 0.976 * | 0.013 | 0.848 * | 1.288 * | 0.994 * | 1.160 * | ↓0.060 | 0.880 |
| + DROPOUT | 1.134 * | 0.209 | 1.161 * | 1.482 * | 1.136 * | 1.321 * | ↑0.134 | 1.074 |
| + INLP | 0.812 * | 0.059 | 0.604 * | 1.407 * | 0.812 * | 1.246 * | ↓0.117 | 0.823 |
| + SENTENCEDEBIAS | 0.755 * | 0.068 | 0.869 * | 1.372 * | 0.774 * | 1.239 * | ↓0.094 | 0.846 |
| RoBERTa large | 0.849 * | 0.170 | -0.237 | 0.900 * | 0.510 * | 1.102 * | ↓0.312 | 0.628 |
| + DYNAMIC PTQ int8 | 0.446 * | 0.218 | -0.368 | 0.423 * | **-0.040** | 0.303 | ↓0.640 | 0.300 |
| Distil RoBERTa | 1.229 * | 0.192 | 0.859 * | 1.504 * | 0.748 * | 1.462 * | ↑0.059 | 0.999 |

Table 8: RACE bias on SEAT dataset. ABWS: angry-black-woman-stereotype. Effect sizes closer to 0 are indicative of less biased model representations. Bold values indicate the best method per test. Statistically significant effect sizes at p < 0.01 are denoted by *. The final column reports the average absolute effect size across all seven race SEAT tests for each model.

| Model | ABWS | ABWS-b | weat3 | weat3b | weat4 | weat5 | weat5b | Avg. Effect |
|---|---|---|---|---|---|---|---|---|
| BERT Base | -0.079 | 0.690 * | 0.778 * | 0.469 * | 0.901 * | 0.887 * | 0.539 * | 0.620 |
| + DYN. PTQ int8 | 0.772 * | 0.425 | 0.835 * | 0.548 * | 0.970 * | 1.076 * | 0.517 * | ↑0.115 0.735 |
| + CDA | 0.231 | 0.619 * | 0.824 * | 0.510 * | 0.896 * | 0.418 * | 0.486 * | ↓0.051 0.569 |
| + DROPOUT | 0.415 * | 0.690 * | 0.698 * | 0.476 * | 0.683 * | 0.417 * | 0.495 * | ↓0.067 0.554 |
| + INLP | 0.295 | 0.565 * | 0.799 * | 0.370 * | 0.976 * | 1.039 * | 0.432 * | ↑0.019 0.639 |
| + SENTDEBIAS | -0.067 | 0.684 * | 0.776 * | 0.451 * | 0.902 * | 0.891 * | 0.513 * | ↓0.008 0.612 |
| BERT Large | -0.219 | 0.953 * | 0.420 * | -0.375 | 0.415 * | 0.890 * | -0.345 | ↓0.104 0.517 |
| + DYN. PTQ int8 | 0.660 * | -0.118 | -0.173 | 0.093 | -0.318 | **0.337** * | 0.364 * | ↓0.305 0.295 |
| Distil BERT | 1.081 * | -0.927 | 0.441 * | 0.202 | 0.358 * | 0.726 * | **-0.076** | ↓0.076 0.544 |
| RoBERTa Base | 0.395 * | 0.159 | -0.114 | **-0.003** | -0.315 | 0.780 * | 0.386 * | 0.307 |
| + DYN. PTQ int8 | 0.660 * | -0.118 | -0.173 | 0.093 | -0.318 | **0.337** * | 0.364 * | ↓0.012 0.295 |
| + CDA | 0.455 * | 0.300 | -0.080 | 0.024 | -0.308 | 0.716 * | 0.371 * | ↑0.015 0.322 |
| + DROPOUT | 0.499 * | 0.392 | -0.162 | 0.044 | -0.367 | 0.841 * | 0.379 * | ↑0.076 0.383 |
| + INLP | 0.222 | 0.445 | 0.354 * | 0.130 | **0.125** | 0.636 * | 0.301 * | ↑0.009 0.316 |
| + SENTDEBIAS | 0.407 * | 0.084 | -0.103 | 0.015 | -0.300 | 0.728 * | 0.274 * | ↓0.034 **0.273** |
| RoBERTa Large | -0.090 | 0.274 | 0.869 * | -0.021 | 0.943 * | 0.767 * | 0.061 | ↑0.125 0.432 |
| + DYN. PTQ int8 | **-0.065** | **-0.014** | 0.587 * | -0.190 | 0.572 * | 0.580 * | -0.173 | ↑0.004 0.312 |
| Distil RoBERTa | 0.774 * | 0.112 | **-0.062** | -0.012 | -0.410 | 0.843 * | 0.456 * | ↑0.074 0.381 |

Table 9: RELIGION bias on SEAT dataset. Effect sizes closer to 0 are indicative of less biased model representations. Bold values indicate the best method per test. Statistically significant effect sizes at p < 0.01 are denoted by *. The final column reports the average absolute effect size across all four religion SEAT tests for each model.

| Model | religion1 | religion1b | religion2 | religion2b | Avg. Abs. Effect. |
|---|---|---|---|---|---|
| BERT Base | 0.744 * | -0.067 | 1.009 * | -0.147 | 0.492 |
| + DYNAMIC PTQ int8 | 0.524 * | -0.171 | 0.689 * | -0.205 | ↓0.095 0.397 |
| + CDA | 0.355 | -0.104 | 0.424 * | -0.474 | ↓0.152 0.339 |
| + DROPOUT | 0.535 * | 0.109 | 0.436 * | -0.428 | ↓0.115 0.377 |
| + INLP | 0.473 * | -0.301 | 0.787 * | -0.280 | ↓0.031 0.460 |
| + SENTENCEDEBIAS | 0.728 * | **0.003** | 0.985 * | 0.038 | ↓0.053 0.439 |
| BERT Large | 0.011 | 0.144 | -0.160 | -0.426 | ↓0.306 0.186 |
| + DYNAMIC PTQ int8 | 0.524 * | -0.171 | 0.689 * | -0.205 | ↓0.095 0.397 |
| Distil BERT | 0.172 | 0.529 * | 0.318 | 0.076 | ↓0.218 0.274 |
| RoBERTa Base | 0.132 | 0.018 | -0.191 | -0.166 | **0.127** |
| + DYNAMIC PTQ int8 | 0.527 * | 0.567 * | **0.079** | 0.020 | ↑0.172 0.298 |
| + CDA | 0.341 | 0.148 | -0.222 | -0.269 | ↑0.119 0.245 |
| + DROPOUT | 0.243 | 0.152 | -0.115 | -0.159 | ↑0.041 0.167 |
| + INLP | -0.309 | -0.347 | -0.191 | -0.135 | ↑0.119 0.246 |
| + SENTENCEDEBIAS | **0.002** | -0.088 | -0.516 | -0.477 | ↑0.144 0.271 |
| RoBERTa Large | -0.163 | -0.685 | -0.158 | -0.542 | ↑0.260 0.387 |
| + DYNAMIC PTQ int8 | 0.117 | -0.292 | 0.293 | **0.015** | ↓0.052 0.179 |
| Distil RoBERTa | 0.490 * | 0.019 | 0.291 | -0.131 | ↑0.106 0.232 |