# OpenReview forum: "Understanding the Effect of Model Compression on Social Bias in Large Language Models"
_EMNLP/2023/Conference — EMNLP 2023 Main_

### Official Review · Reviewer_KtY1 · 2023-07-25

**Typos Grammar Style And Presentation Improvements:** 1. Explain what is PTQ when it first …
**Soundness:** 3

**Excitement:**

3: Ambivalent: It has merits (e.g., it reports state-of-the-art results, the idea is nice), but there are key weaknesses (e.g., it describes incremental work), and it can significantly benefit from another round of revision. However, I won't object to accepting it if my co-reviewers champion it.

**Paper Topic And Main Contributions:**

This paper examines the effect of model compression techniques like quantization and knowledge distillation on social biases in large language models (LLMs).

**Key points:**

- This paper evaluates compressed versions of BERT, RoBERTa, and Pythia LLMs on benchmark datasets measuring gender, race, and religion biases.

- The results show that compression via quantization and distillation tends to reduce social bias while maintaining language modeling performance. The reductions are comparable to specifically designed debiasing techniques.

- Longer pretraining and larger uncompressed models lead to higher social bias.

**Questions For The Authors:**

1. In Table 1, the Stereotype scores closer to 50% indicate a less biased model. For the RELIGION column, why you highlight the 40.95 which is not the result that is the closest to 50%? This makes me quite confused.

**Reasons To Accept:**

1. The paper explores an underexamined interplay between model compression and social bias reduction in LLMs. This is an interesting new direction for bias analysis.

2. The experiments systematically evaluate various compression methods on multiple model architectures. The results clearly demonstrate compression can reduce biases to a certain degree.

3. The writing clearly explains the background, experiments, results, and implications. The visualizations effectively summarize key trends.

Overall, this paper makes a solid contribution to an important emerging topic with thoughtful experiments and analysis. The presentation and writing quality are publication-worthy.

**Reasons To Reject:**

1. Limited technical contributions. The compression techniques evaluated are standard existing methods like quantization and distillation. The debiasing baselines are also from prior work. There is little technical innovation.

2. Limited datasets and models. The bias benchmarks only assess gender, race, and religion. Other important biases and datasets are not measured. Also missing are assessments on state-of-the-art generative models like GPT.

3. Writing logic needs improvement. Some parts, like introducing debiasing baselines in the results, make the flow confusing.

**Reproducibility:**

4: Could mostly reproduce the results, but there may be some variation because of sample variance or minor variations in their interpretation of the protocol or method.

**Reviewer Confidence:**

3: Pretty sure, but there's a chance I missed something. Although I have a good feel for this area in general, I did not carefully check the paper's details, e.g., the math, experimental design, or novelty.

---

> ### Author Rebuttal · Authors · 2023-08-29
>
> Thank you for taking the time to carefully review our paper. We appreciate the opportunity to address your concerns and present our case for the acceptance of this paper.
>
> We are pleased that you found merit in our paper and its contributions to the field. Our work aims to explore the potential of using out-of-the-box compression methods, which are easily accessible and are the first line of techniques for deployment in production environments. Therefore, we focused on simple compression techniques, whose effects on social bias are underexamined.
> We recognize the simplicity of the techniques used in this work, which we believe to be a strength of the paper, demonstrating that out-of-the-box compression methods can already help mitigate social bias. We also acknowledge that this work examines a subset of all social biases, as stated in the Limitations section of our paper. Yet, we feel that we experimented with a wide enough variety of biases, techniques and models to demonstrate the promise of methods combining compression and social bias mitigation sufficient to demonstrate a clear trend and inspire future research.
>
> We want to highlight that our experiments were designed to include both encoder and decoder-only variations of LLMs. We chose the Pythia model family to represent state-of-the-art decoder-only models since it uses an open-source architecture based on GPT-NeoX [1].
> While we are also curious about the performance of other state-of-the-art decoder-only LLMs such as OpenAI’s GPT models, we note that it is currently impossible to perform this study on those models due to the model parameters not being shared, hence we used a close open-source alternative., We believe there is significant value in embracing open-source initiatives that allow us to peek into the performance during pretraining, which is a perspective currently beyond reach with OpenAI’s GPT models.
>
> We noticed that we got an average reproducibility score, but could not find improvement points with respect to reproducibility. We will be happy to revise this either on the main paper, or appendix, and we will release the modifications that we made to the BiasBench codebase.
>
> Finally, we want to thank you for your technical writing improvements, which we will incorporate into the final version of our paper.
>
> [1] Black, Biderman, Hallahan, Anthony, Gao, Golding, He, Leahy, McDonell, Phang, Pieler, Prashanth, Purohit, Reynolds, Tow, Wang, and Weinbach. "GPT-NeoX-20B: An Open-Source Autoregressive Language Model." In Proceedings of the ACL Workshop on Challenges & Perspectives in Creating Large Language Models. 2022.
>
>
> **QA Section**
>
> Q1: "In Table 1, the Stereotype scores closer to 50% indicate a less biased model. For the RELIGION column, why you highlight the 40.95 which is not the result that is the closest to 50%? This makes me quite confused."
>
> A1: Thank you for pointing this out. You are correct, and we apologize for the confusion. The intended emphasis was on the Dynamic PTQ int8 cell with the value of 49.52, which is indeed the value closest to 50%. We will amend this for the final version of the paper.

---

### Official Review · Reviewer_nH78 · 2023-07-28

**Soundness:** 4

**Excitement:**

3: Ambivalent: It has merits (e.g., it reports state-of-the-art results, the idea is nice), but there are key weaknesses (e.g., it describes incremental work), and it can significantly benefit from another round of revision. However, I won't object to accepting it if my co-reviewers champion it.

**Missing References:**

1. Webster, K., Wang, X., Tenney, I., Beutel, A., Pitler, E., Pavlick, E., Chen, J., Chi, E. and Petrov, S., 2020. Measuring and reducing gendered correlations in pre-trained models. arXiv preprint arXiv:2010.06032.
2. Kaneko, M. and Bollegala, D., 2022, June. Unmasking the mask–evaluating social biases in masked language models. In Proceedings of the AAAI Conference on Artificial Intelligence, pp. 11954-11962.

**Paper Topic And Main Contributions:**

In this paper, the authors study the effects of model compression approaches (i.e., quantization and knowledge distillation) on social bias measures in large language models (LLMs). Their findings show that longer pretraining and larger models tend to exhibit higher levels of social bias. Moreover, the study demonstrates that compression approaches, such as quantization and knowledge distillation, have the potential to reduce social bias in LLMs.

**Questions For The Authors:**

1. Given that CrowS-Pairs and StereoSet contain 9 and 4 types of social bias, respectively, I wonder why you only consider the three identity categories rather than all bias types that are included in the CrowS-Pairs and StereoSet?

2. There are other debiasing methods widely used for debiasing LMs, such as Counterfactual Data Augmentation debiasing (CDA; Webster et al., 2020), dropout debiasing (DO; Webster et al., 2020) and Token-level debiasing (AT), did you compare the results of using compression methods with them?

**Reasons To Accept:**

This paper presents a comprehensive investigation into the impact of model compression techniques, specifically quantization and knowledge distillation, on social bias measures in large language models (LLMs). The study provides substantial evidence supporting the observation that longer pretraining durations and larger model sizes are associated with higher levels of social bias in LLMs. This work sheds light on the possibility of mitigating social bias in LLMs through compression techniques, even without explicitly engaging in the process of debasing. The paper is easy to follow.

**Reasons To Reject:**

Further investigation and analysis are necessary especially to ascertain the presence and persistence of social biases in the student model after the training process. Kaneko et al. (2022) showed that LMs debiased using different methods still re-learn social bias during finetuning on downstream tasks. It is not clear whether the social biases present in DistillBERT are re-learned during the training of the student model.

Furthermore, it should be noted that the statement made in lines 258-259, claiming that "longer pretraining and larger models lead to more socially biased models", is drawn from observations solely concerning Pythia models (results shown in Figure 1). However, this statement does not hold true when comparing BERT base and BERT large models, as demonstrated in Table 1. Specifically, the analysis reveals that BERT large achieves lower bias scores on both gender and race biases.

**Reproducibility:**

4: Could mostly reproduce the results, but there may be some variation because of sample variance or minor variations in their interpretation of the protocol or method.

**Reviewer Confidence:**

3: Pretty sure, but there's a chance I missed something. Although I have a good feel for this area in general, I did not carefully check the paper's details, e.g., the math, experimental design, or novelty.

**Typos Grammar Style And Presentation Improvements:**

It would be better to mention the full name of "PTQ" (in line 85) when first mentioned to ensure clarity and proper identification.

---

> ### Author Rebuttal · Authors · 2023-08-29
>
> Thank you for taking the time to carefully review our paper. We appreciate the opportunity to address your concerns and present our case for the acceptance of this paper.
>
> We are pleased that you found merit in our paper and its contributions to the field. We acknowledge that there is more analysis to be done on the impacts of distillation on social biases. However, to achieve this analysis we would need to engage in the task of creating our own version of the distilled BERT and RoBERTa models, which escapes our out-of-the-box premise. The disclosure of intermediary checkpoints for the distilled student versions would enable a swift study of bias evolution for distilled models. Nevertheless, our results allow us to understand that the pre-trained distilled models tend to significantly decrease bias. We agree that a logical next step would be to analyze the effects of bias during the distillation of the pre-training process.
> We agree that the study of re-learned biases during finetuning is an interesting and complementary experiment to this work. However, for this short paper, we focus the scope of our work on the pre-trained models, before finetuning.
> With the simplicity of the techniques used in this work, we hope to inspire future research into the study of more elaborate compression methods across a variety of LLMs, while also examining social bias.
>
> Tables 1,4,7,8, and 9 explicitly compare intra-model differences i.e. Base, Large, and Distilled through the absolute difference alongside the table values. We can also compare the inter-model performance across those model sizes by performing a rank analysis. This involves determining the ranking (1st, 2nd, or 3rd) of models with lower biases. We can observe an average rank order, according to the table below: Distil < Base < Large.
>
> Model   |  Avg. Rank
>
> Distill    |     1.56
>
> Base    |     2.17
>
> Large   |     2.28
>
> We observed in our experiments that some biases will not benefit from the regularization effects of compression methods. However, the overall trend is that larger models and longer training can lead to higher biases.
> Our choice of model parameters across the Pythia family was also to observe the effects of similar-sized models to Large (Pythia-410M), Base (Pythia-160M), and Distilled (Pythia-70M). This exercise needs to be done with caution, as we are comparing encoder models with decoder-only, where encoders benefit from bidirectionality while decoders do not. We can observe a consistent pattern in the line plots where the prevailing order from more biased models to less biased models is 410M (Green) > 160M (Yellow) > 70M (Blue). Moreover, PTQ tends to lower the bias during some pretraining parts across these 3 model sizes.
>
> Finally, we want to thank you for your technical writing improvements, which we will incorporate into the final version of our paper.
>
> **QA Section**
>
>
>
> Q1: "Given that CrowS-Pairs and StereoSet contain 9 and 4 types of social bias, respectively, I wonder why you only consider the three identity categories rather than all bias types that are included in the CrowS-Pairs and StereoSet?"
>
> A1: We limited our analysis to 3 types of social bias, influenced by building on top of the BiasBench benchmark [1]. This benchmark focuses on the 3 types of social biases that are common across datasets and allow for a high-level comparison. We acknowledge in the Limitations section that this is a limited representation of social bias that needs to be expanded in future work.
>
> Q2: "There are other debiasing methods widely used for debiasing LMs, such as Counterfactual Data Augmentation debiasing (CDA; Webster et al., 2020), dropout debiasing (DO; Webster et al., 2020) and Token-level debiasing (AT), did you compare the results of using compression methods with them?"
>
> A2: We indirectly compared our results with CDA (Webster et al., 2020), and dropout debiasing (DO; Webster et al., 2020). These two methods are baselines of the BiasBench (Meade et al. 2022) paper, which forms the base for our evaluation. Due to space constraints, we report INLP, Self-Debias, and SentenceDebias baselines as they are, on average, more performant than CDA and DO when applied to BERT and RoBERTa. These can be included in the final version of our paper for clarity, as these results were already reported in the BiasBench paper. With respect to the Token-level debiasing (AT), we did not compare against this debiasing technique as it happens at fine-tuning time and our work focuses on the effects of social-bias using out-of-the-box pre-trained models.
>
> [1] - Nicholas Meade, Elinor Poole-Dayan, Siva Reddy
> An Empirical Survey of the Effectiveness of Debiasing Techniques for Pre-trained Language Models. ACL (1) 2022: 1878-1898

---

### Official Review · Reviewer_az8p · 2023-08-05

**Soundness:** 3

**Excitement:**

3: Ambivalent: It has merits (e.g., it reports state-of-the-art results, the idea is nice), but there are key weaknesses (e.g., it describes incremental work), and it can significantly benefit from another round of revision. However, I won't object to accepting it if my co-reviewers champion it.

**Paper Topic And Main Contributions:**

The paper empirically analyzes the effect of model compression on the social bias of pretrained language models: knowledge distillation and dynamic post-training quantization (PTQ). In the experiment, social bias is evaluated with three intrinsic bias benchmarks (CrowS-Pairs, StereoSet, SEAT) that cover three types of bias (gender, race, religion). The experimented models include BERT-base/large, RoBERTa-base/large, and Pythia models (decoder-only) with various sizes.

**Questions For The Authors:**

- A. Table 2-3 in the main text are captioned as SteroSet dataset with Pythia and Pythia-int8. Table 5-6 in the appendix have seemingly the same captions but with different values. Please clarify this.

- B. Question on DistillBERT bias score (Table 1: CrowS data): 51.15, 46.99, 58.10 (gender, race, religion). The score is notably different from that of a recent paper [2]: 56.87, 60.97, 66.67 (see their Table 2). Could you explain the possible reasons for the discrepancy?

[2] https://aclanthology.org/2023.acl-long.878.pdf (I understand the paper was published shortly before the EMNLP deadline and thus there is no novelty concern.)

**Reasons To Accept:**

- With the help of recent Pythia model checkpoints, the paper investigates bias progression along with various model sizes and training steps.

- The experiment validates the positive effect of PTQ on CrowS and SteroSet. (1) It consistently reduces bias for BERT-base/large and RoBERTa-base/large for CrowS and SteroSet (Table 1, Appendix Table 4). (2) For SteroSet, when selecting models based on the best LM score, it reduces bias of Pythia (Table 2, 3). It additionally shows best LM score of PTQ occurs early in the training progress. (3) When comparing models with the same training steps, PTQ usually reduces biases (Appendix Figure 2-7).

**Reasons To Reject:**

- The effect of knowledge distillation is not sufficiently studied. (1) The conclusion is based only on the comparison between BERT-base vs. DistilBERT and RoBERTa-base vs. DistilRoBERTa. (2) The conclusion that KD reduces bias is mostly true for CrowS (Table 1) and  SteroSet (Appendix Table 4) datasets (although the religion bias sometimes increases). However, the SEAT results have a mixed trend (Appendix Table 7-9): DistillBERT reduces bias while DistilRoBERTa increases bias, in terms of average effect.

- As acknowledged above, PTQ has positive results on CrowS and SteroSet. However, the result on SEAT is somewhat negative.
(1) PTQ effect of BERT and RoBERTa is ambivalent (Appendix Table 7-9): BERT-base (gender -0.158, race +0.115, religion -0.095), bert-large (+0.226, -0.305, -0.095), roberta-base (-0.406, -0.012, +0.172), roberta-large (-0.64, +0.004, +0.052).
(2) For PTQ of Pythia (Appendix Figure 9-11), bias often increases for moderate-size models (i.e., >=410M), when comparing models with the same training steps. That is, when overlapping the left and right curves, the PTQ curve often occurs above the original curve.

- In summary, both methods reduce bias in CrowS and SteroSet datasets. However, the appendix shows a different trend with the SEAT dataset. Considering CrowS and SteroSet datasets are under question [1], the inconsistent result from SEAT raises more concerns.

[1] https://aclanthology.org/2021.acl-long.81.pdf

**Reproducibility:**

3: Could reproduce the results with some difficulty. The settings of parameters are underspecified or subjectively determined; the training/evaluation data are not widely available.

**Reviewer Confidence:**

3: Pretty sure, but there's a chance I missed something. Although I have a good feel for this area in general, I did not carefully check the paper's details, e.g., the math, experimental design, or novelty.

**Typos Grammar Style And Presentation Improvements:**

- Table 1 & 4. The side numbers (relative change) and boldness are sometimes misleading. Since the optimal score is 50 for CrowS &  SteroSet, the improvement should be based on absolute deviation. For instance, in Table 1 religion column, roberta-base+ptq should have side number: abs(49.52-50)-abs(60.95-50)=-10.47, and this cell should be bolded since this is the least biased one (closest to 50). This is the practice used in the prior work [3]; see their Table 4 GPT2+SentenceDebias.

- Table 5 caption: steps of "int8" model, LM score of "original" model. Table 6 caption: steps of "original" model LM score of "int8" model. Possibly a mix-up.

[3] https://aclanthology.org/2022.acl-long.132.pdf

---

> ### Author Rebuttal · Authors · 2023-08-29
>
> Thank you for taking the time to carefully review our paper. We appreciate the opportunity to address your concerns and present our case for the acceptance of this paper.
>
> We are pleased that you found merit in our paper and its contributions to the field, namely: valuable insights into the utilization of out-of-the-box compression methods as potential solutions for mitigating social bias.  The effects of compression on LLMs' social bias are still underexplored, and we focused particularly on studying the social biases of these models while in their pre-trained state.
>
> We acknowledge that the results from the SEAT dataset show a mixed narrative, though we disagree that mixed results should be a reason for rejection. With an extra page of content, we can add more discussion of these results into the paper. Notably, RoBERTa Base/Large achieves a competitive lower average bias that the distilled version struggles to enhance (Tables 7 to 9). However, the trend that the distilled version is better holds true for BERT. It's important to note that RoBERTa is a better-trained version of BERT, with a longer training duration over a larger dataset. We hypothesize that the SEAT task is more complex and benefits from models with more representativeness, thus making it more challenging to improve on social bias with out-of-the-box compression methods, which can crudely reduce the model's expressiveness.
>
> The ambivalence of results in applying the studied out-of-the-box compression algorithms on the SEAT dataset is something to be noted. We argue that even though these straightforward methods may not enhance every social bias or perform consistently across all models, our work demonstrates their potential as effective strategies for swiftly reducing social bias. We do not anticipate these methods to consistently outperform debiasing approaches, but we believe that achieving similar performance to debiasing methods will inspire the community to expand on this line of research, for example by exploring methodology in the intersection of compression and debiasing.
>
> We noticed that we got an average reproducibility score, but could not find improvement points with respect to reproducibility. We will be happy to revise this either on the main paper, or appendix, and we will release the modifications that we made to the BiasBench codebase.
>
> Finally, we want to thank you for your technical writing improvements, which we will incorporate into the final version of our paper.
>
> **QA Section**
>
> Q1: "Table 2-3 in the main text are captioned as SteroSet dataset with Pythia and Pythia-int8. Table 5-6 in the appendix have seemingly the same captions but with different values. Please clarify this."
>
> A1: Thank you for raising this point, as it shows that a rephrasing of the captions is required. In Tables 2 and 3 we analyze the three types of bias at the point where the LM score reaches its peak, comparing the unquantized and quantized models.
> Moving on to Tables 5 and 6, we utilize the frozen step numbers determined in Tables 2 and 3 to examine both the LM score performance and social bias performance in the opposite model. In other words, we evaluate the performance of the original/quantized model at the point where the previous one had shown its peak. For instance, referring to Table 2, we note that for the unquantized model with 70M parameters, the highest LM score is 89.2 at step 21K, accompanied by its respective biases. Now, turning to Table 5, we observe that at the same step, 21K, which yielded the best result for the unquantized model, the LM score for the quantized model is 87.7.
>
> Q2: "Question on DistillBERT bias score (Table 1: CrowS data): 51.15, 46.99, 58.10 (gender, race, religion). The score is notably different from that of a recent paper [2]: 56.87, 60.97, 66.67 (see their Table 2). Could you explain the possible reasons for the discrepancy?
> [2] https://aclanthology.org/2023.acl-long.878.pdf (I understand the paper was published shortly before the EMNLP deadline and thus there is no novelty concern.)"
>
> A2: Thank you for pointing out this very interesting work. Their DistilBERT results are indeed very different from ours, unfortunately, it does not seem that they open-sourced their codebase, and included implementation details such as the used seed to generate such results. The utilization of different seeds has a big performance impact as shown by Sellam et al. (Figure 4) [3]. We think this could be one of the primary reasons for the discrepancy of our results with [2]. We utilize the BiasBench codebase to reproduce results and perform our compression experiments, without further details from [2] we cannot say with certainty why the results differ.
> [3] https://arxiv.org/pdf/2106.16163.pdf

---

### Meta-Review · Area_Chair_bMKe · 2023-09-24

**Recommendation:** 4

**Metareview:**

This paper studies how the compression of LLMs affects their social biases. This kind of study is very timely as efficient versions of LLMs are becoming more widespread and may affect users in unpredictable ways. An interesting aspect of this study is that it ablates its results over model size and training steps, by comparing different model checkpoints. The findings are non-trivial, as the Authors find that quantisation and distillation can reduce bias. However, the results on the SEAT dataset remain mixed. Moreover, some conclusions regarding the effect of model size based on Pythia seem to be contradicted by BERT-style models. Both of these points warrant further discussion on possible explanations, but overall I recommend to accept this paper.

---

### Decision · Program_Chairs · 2023-10-07

**Decision:**

Accept-Main

**Comment:**

This paper studies how the compression of LLMs affects their social biases. This kind of study is very timely as efficient versions of LLMs are becoming more widespread and may affect users in unpredictable ways. An interesting aspect of this study is that it ablates its results over model size and training steps, by comparing different model checkpoints. The findings are non-trivial, as the Authors find that quantisation and distillation can reduce bias. However, the results on the SEAT dataset remain mixed. Moreover, some conclusions regarding the effect of model size based on Pythia seem to be contradicted by BERT-style models. Both of these points warrant further discussion on possible explanations, but overall I recommend to accept this paper.